# A Co-Sputtering Process Optimization for the Preparation of FeGaB Alloy Magnetostrictive Thin Films

**DOI:** 10.3390/nano13222948

**Published:** 2023-11-15

**Authors:** Qijing Lin, Zelin Wang, Qingzhi Meng, Qi Mao, Dan Xian, Bian Tian

**Affiliations:** 1State Key Laboratory of Mechanical Manufacturing Systems Engineering, Xi’an Jiaotong University, Xi’an 710049, China; qjlin2015@xjtu.edu.cn (Q.L.); wzl15086927209@stu.xjtu.edu.cn (Z.W.); danxian@xjtu.edu.cn (D.X.); t.b12@mail.xjtu.edu.cn (B.T.); 2School of Mechanical and Manufacturing Engineering, Xiamen Institute of Technology, Xiamen 361021, China; 3Shandong Laboratory of Yantai Advanced Materials and Green Manufacturing, Yantai 265503, China; 4Xi’an Jiaotong University (Yantai) Research Institute for Intelligent Sensing Technology and System, Xi’an Jiaotong University, Xi’an 710049, China

**Keywords:** co-sputtering, magnetostrictive thin films, soft magnetic performance, coercivity, magnetoelectric sensors

## Abstract

A co-sputtering process for the deposition of Fe_0.8_Ga_0.2_B alloy magnetostrictive thin films is studied in this paper. The soft magnetic performance of Fe_0.8_Ga_0.2_B thin films is modulated by the direct-current (DC) sputtering power of an FeGa target and the radio-frequency (RF) sputtering power of a B target. Characterization results show that the prepared Fe_0.8_Ga_0.2_B films are amorphous with uniform thickness and low coercivity. With increasing FeGa DC sputtering power, coercivity raises, resulting from the enhancement of magnetism and grain growth. On the other hand, when the RF sputtering power of the B target increases, the coercivity decreases first and then increases because of the conversion of the films from a crystalline to an amorphous state. The lowest coercivity of 7.51 Oe is finally obtained with the sputtering power of 20 W for the FeGa target and 60 W for the B target. Potentially, this optimization provides a simple way for improving the magnetoelectric coefficient of magnetoelectric composite materials and the sensitivity of magnetoelectric sensors.

## 1. Introduction

During the past decades, it has been discovered that the magnetoelectric coefficient of composite materials is much higher than that of single-phase materials. Moreover, these materials can directly convert magnetic signals into electrical signals even under passive conditions. Therefore, magnetoelectric composite materials can be used as sensitive elements in magnetoelectric sensors and are widely applied in the fields of microwave communication [1,2,3], band-pass filters [4,5,6], phase shifters [7,8,9], etc.

Magnetoelectric composite materials are multi-phase composite materials that combine magnetostrictive layers and piezoelectric layers. In order to obtain a high magneto-electric coefficient, it is necessary to choose materials with greater magnetostrictive coefficients. FeGaB alloy is a type of material with a high magnetostrictive coefficient, low saturation magnetic field, and excellent soft magnetic performance. By optimizing the deposition process, the soft magnetic and magnetostrictive performance of FeGaB films can be improved, thereby enhancing the sensitivity of magnetoelectric sensors.

Currently, methods for preparing FeGaB films include pulsed laser deposition (PLD) [10,11,12], the sol-gel method [13,14,15], and magnetron sputtering deposition [16,17,18]. Films prepared by PLD technology exhibit controllable thickness and smooth surface morphology, but the coating area is small and the deposition rate is low. The sol-gel method allows for easy and uniform doping of trace elements at the molecular level, but the entire sol-gel process takes a long time and the gel contains a large number of micropores, which will release a lot of gas and organic compounds during the drying process, leading to a shrinkage of the composite films. Magnetron sputtering deposition has the advantages of a large coating area, slow heating of the substrate, good adhesion, and low cost, so is an excellent choice for the preparation of magnetostrictive materials. There are two approaches employing magnetron sputtering using (i) a single cathode composed of FeGaB alloy or (ii) co-sputtering [19,20] with two FeGa and B cathodes. The first method requires high accuracy for the element composition and uniformity of the target, and it is difficult to control the element composition by sputtering parameters, resulting in low target utilization. By controlling the sputtering parameters of different targets, the co-sputtering process can regulate the film composition, making the films more uniform and optimizing the soft magnetic performance. Meanwhile, co-sputtering has higher target utilization, better flexibility, and operability.

In this study, a co-sputtering process with FeGa alloy and B targets was carried out for the deposition of FeGaB magnetostrictive films, and the sputtering power was optimized for the improvement of soft magnetic performance. The composition, surface morphology, and coercivity of the Fe_0.8_Ga_0.2_B films under different sputtering process parameters were characterized and analyzed by using X-ray diffraction (XRD, D8 ADVANCE A25, Saarbrucken, Germany), atom force microscopy (AFM, INNOVA, Billerica, MA, USA), and vibrating sample magnetometer (VSM, Lakeshore 7404, Great Barrington, MA, USA) methods. The influence of sputtering process parameters on the soft magnetic performance of the FeGaB films was systematically analyzed, which proposed a potential way for the enhancement of magnetoelectric (ME) sensor sensitivity.

## 2. Growth Process of Fe_0.8_Ga_0.2_B Magnetostrictive Films

The composition ratio and structure parameters play a crucial role in determining the magnetostrictive properties of FeGaB [21] thin films, which can be regulated by process parameters. A cross-sectional diagram of the composite film is shown in Figure 1. The substrate is single-crystal silicon with a thickness of 500 μm, on which is 500 nm Mo, 1 μm Al_0.8_Sc_0.2_N, and Fe_0.8_Ga_0.2_B magnetostrictive film in sequence. As the composite film is used for the preparation of an ME sensor, Mo acts as the bottom electrode of the sensor, Al_0.8_Sc_0.2_N is the piezoelectric film, and Fe_0.8_Ga_0.2_B thin film is the magnetostrictive layer.

The Mo metal film is deposited by the magnetron sputtering process, and the Al_0.8_Sc_0.2_N [22] piezoelectric film is prepared by the co-sputtering method using Al and Sc targets in a nitrogen environment at the temperature of 350 °C. For the Fe_0.8_Ga_0.2_B magnetostrictive film, a magnetron sputtering system with Fe_0.8_Ga_0.2_ alloy as the DC target and pure B material as the RF target is carried out. The coercivity and deposition rate of Fe_0.8_Ga_0.2_B can be adjusted by changing the sputtering power parameters of the targets, in order to obtain an Fe_0.8_Ga_0.2_B thin film with lower coercivity and better uniformity.

## 3. Characterization Method of Fe_0.8_Ga_0.2_B Magnetostrictive Films

After the growth of Fe_0.8_Ga_0.2_B magnetostrictive films, the composition, surface morphology, and coercivity of the films are characterized by the XRD, AFM, and VSM methods.

Through the XRD scanning spectrum, the types and composition of the films can be identified, and the quality (e.g., crystallinity and grain size) of the films can also be analyzed by the intensity of the diffraction peaks. As the major peaks of the films like Al_0.8_Sc_0.2_N, Fe_0.8_Ga_0.2_B, etc. are in the range of 36° to 65°, the scanning range is set as 30° to 70°.

By AFM measurement, the surface morphology and root mean square (RMS) roughness value of the films can be obtained for the evaluation of the deposition uniformity. Herein, a scanning area of 5 μm × 5 μm is used because it can not only reflect the relatively complete uniformity of the film, but also ensures high efficiency.

The magnetic hysteresis loops of the samples can be measured by a vibrating sample magnetometer (VSM). By applying an external magnetic field from −1000 Oe to 1000 Oe and repeating this process, the magnetic hysteresis loop of the Fe_0.8_Ga_0.2_B film is obtained through measuring its magnetization intensity. The coercivity is defined as the external magnetic field applied on the films when the internal magnetization intensity of the films is zero. Thus, through the analysis of VSM curves, the coercivity of the films can be obtained. A lower coercivity of a film means better soft magnetic properties and a greater sensitivity of an ME sensor, so by comparing the coercivity of different samples, the optimal process parameters can be achieved.

## 4. Results and Discussion

A Fe_0.8_Ga_0.2_B alloy magnetostrictive thin film is initially deposited at sputtering conditions of DC power of 30 W and RF power of 46 W. As illustrated in Figure 2, the surface topography of the composite film has been studied by atomic force microscopy (AFM), which shows that the root mean square (RMS) roughness value is below 9.8 nm. Hence, good uniformity of thin films with columnar growth has been obtained from sputtering.

Figure 3 shows the XRD pattern of the substrate surface after sputtering Fe_0.8_Ga_0.2_B with a DC power of 30 W and an RF power of 46 W. A diffraction peak at the degree of 36° shows that the Al_0.8_Sc_0.2_N piezoelectric film exhibits a typical (002) direction crystal and superior piezoelectric performance. On the other hand, the sputtered Fe_0.8_Ga_0.2_B film shows a more pronounced diffraction peak at 44° [21], which means that the magnetostrictive film has large grains in the crystal and great coercivity. Fe_0.8_Ga_0.2_BO_3_ is possibly the natural oxide of Fe_0.8_Ga_0.2_B thin film in air, and Fe (200) is the iron elemental in Fe_0.8_Ga_0.2_B film. Due to the natural quality of the crystalline in the film, under the external magnetic field it is difficult to achieve the rotation of magnetic domains in a high crystallinity film. So it exhibits a higher coercivity and poorer soft magnetic properties. By optimizing the sputtering power and exploring the process conditions, the best soft magnetic performance of the Fe_0.8_Ga_0.2_B film can be achieved.

Figure 4 shows the variation curve of the thickness of Fe_0.8_Ga_0.2_B thin film with Fe_0.8_Ga_0.2_ target power. Maintaining the power of target B at 46 W, the thickness of the deposited Fe_0.8_Ga_0.2_B film changes from 247.2 nm to 318.2 nm with an increase in the FeGa alloy target power from 20 W to 40 W. There is a linear relationship between the power of the FeGa target and the thickness of the thin film. While keeping the power of the FeGa target at 30 W, the thickness of Fe_0.8_Ga_0.2_B film only ranges from 247.2 nm to 250.1 nm with an increase in the power of the B target from 46 W to 67 W.

It can be seen that the sputtering power of FeGa alloy targets can significantly change the deposition rate. Increasing the DC sputtering power results in more Ar+ ions being produced, which leads to higher energy for the atoms on the target surface. As a result, more target atoms are ejected from the target surface and deposited onto the substrate. However, excessively high sputtering power will lead to high coercivity in the Fe_0.8_Ga_0.2_B film, while low sputtering power can result in weak glow discharge during the sputtering process, which is not conducive to maintaining the sputtering process. The power of the B target has a minor influence on the deposition rate of the Fe_0.8_Ga_0.2_B film. In the sputtering process of the Fe_0.8_Ga_0.2_B film, the sputtering efficiency of the RF target is much lower than that of the DC target, so the sputtering yield of B atoms is relatively small and has a minor effect on the deposition rate.

Figure 5 is the XRD image of Fe_0.8_Ga_0.2_B thin films corresponding to different sputtering powers of FeGa targets. The B target power remains at 46 W, and when the FeGa sputtering power changes, strong diffraction peaks appear at both 25 W, 30 W, and 40 W. The grain size of the thin film can be calculated using Scherrer’s formula:(1)$$D=\frac{\kappa \lambda }{\beta cos(\theta )}$$
where *κ* is the Scherrer constant, *λ* is the wavelength of the X-ray, *β* is the full width at half maximum (FWHM) of the diffraction peak, and *θ* is the Bragg diffraction angle. The grain size of the Fe_0.8_Ga_0.2_B thin film with different powers of FeGa target is displayed in Table 1. When the power of the FeGa target is 40 W, the FWHM of the diffraction peak of the film is 0.496° and the grain size of the Fe_0.8_Ga_0.2_B thin film obtained by sputtering at this power is 11.8 nm. When the power of the FeGa target is 30 W, the FWHM of the diffraction peak of the film is 0.715°, and the grain size of the Fe_0.8_Ga_0.2_B thin film obtained by sputtering at this power is 8.2 nm. When the power of the FeGa target is 25 W, the FWHM of the diffraction peak for the deposited film is 0.840° and the grain size of the Fe_0.8_Ga_0.2_B film is 7.0 nm. As the power of the FeGa target decreases to 20 W, the diffraction peak tends to flatten, indicating that as the B content in the film components relatively increases, the grain size of the film is refined, and the Fe_0.8_Ga_0.2_B thin film becomes amorphous; Boron atoms are expected to distribute around the boundaries of grains and suppress their growth.

Figure 6 shows the magnetic hysteresis loops of thin films formed at different sputtering powers of FeGa alloy targets, which are measured by a vibrating sample magnetometer (VSM). It is shown that the coercivity of the thin film raises with an increase in FeGa alloy target sputtering power, and the minimum coercivity is 17.72 Oe. Therefore, when there is a specific ratio between FeGa alloy and B target materials, a better soft magnetic property can be achieved. By comparing with the XRD graph, it can be observed that the smaller the FWHM of the diffraction peaks in the Fe_0.8_Ga_0.2_B thin film, the larger the coercivity will be. When the B element is introduced into the FeGa thin film, the Fe_0.8_Ga_0.2_B thin film transforms into an amorphous phase, leading to an increase in FWHM, which means a small coercivity. This is beneficial for eliminating magnetic crystalline anisotropy, thereby improving the soft magnetic performance of the Fe_0.8_Ga_0.2_B thin film.

Figure 7 shows the XRD spectra of Fe_0.8_Ga_0.2_B thin film at different powers for B target sputtering. The FeGa target power is set at 20 W. At a B target sputtering power of 46 W, a pronounced FeGa(110) diffraction peak appears at around 44°. From the XRD spectra, it can be observed that as the B target power increases, the intensity of this diffraction peak decreases gradually and tends to flatten. This suggests that the increased B target power leads to a relatively higher B content in the Fe_0.8_Ga_0.2_B film, causing it to transform into an amorphous state. The FeGa lattice becomes more disordered, and B atoms fill the gaps within the FeGa lattice, resulting in further refinement of the FeGa grains and the gradual disappearance of sub-grain boundaries. This decrease in grain size reduces the resistance to magnetization domain rotation under external magnetic fields. When the B target power reaches 67 W, the diffraction peak becomes slightly obvious. Moreover, it is inferred that as the B target sputtering power continues to increase, the intensity of the diffraction peak will also increase. This could be attributed to the excessively high B target power leading to a high quantity of B atoms in the film, and as a result, the film deposition becomes non-uniform and discontinuous, hindering the homogeneous distribution of boron atoms within the Fe_0.8_Ga_0.2_B film. Consequently, the film surface becomes uneven, and the overall film quality deteriorates, resulting in larger crystallinity.

In order to enhance the sensitivity of an ME sensor for the detection of weak magnetic fields, it is necessary to select Fe_0.8_Ga_0.2_B films with lower coercivity. The coercivity of the Fe_0.8_Ga_0.2_B films deposited at different B powers is measured by a VSM, as shown in Figure 8. As the B target power increases, the coercivity of the film first decreases and then increases. The minimum coercivity of 7.51 Oe is achieved. The coercivity values match well with the diffraction peaks in the XRD graph in Figure 6, indicating that the amorphous nature of the Fe_0.8_Ga_0.2_B film is an important factor for its excellent soft magnetic properties.

## 5. Conclusions

The effects of sputtering power from FeGa and B targets on the deposition rate, soft magnetic performance, and coercivity of Fe_0.8_Ga_0.2_B thin films were investigated using the magnetron co-sputtering method. The prepared Fe_0.8_Ga_0.2_B thin films’ thickness raises sharply with increasing FeGa target sputtering power, while it remains almost constant with variations in the B target power. The XRD and VSM results show that as the power of FeGa targets decreases from 40 to 20 W, the Fe_0.8_Ga_0.2_B thin film becomes amorphous and the coercivity decreases. However, with increasing B sputtering power, the B content in the Fe_0.8_Ga_0.2_B films increases, and the coercivity initially decreases and then increases. When the B sputtering power increases to 60 W, the coercivity of the Fe_0.8_Ga_0.2_B film decreases to 7.51 Oe, and further increasing the sputtering power leads to an increase in coercivity. In general, an FeGa power of 20 W and a B sputtering power of 60 W resulted in Fe_0.8_Ga_0.2_B films with low coercivity and a high magnetostriction coefficient, providing potential for the fabrication of high-frequency magnetoelectric sensors with high magnetic coupling coefficients and sensitivity.

## Figures and Tables

**Figure 1 nanomaterials-13-02948-f001:**
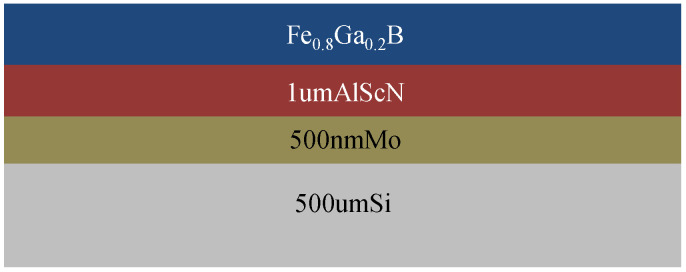
A diagram of the cross-section of the composite film.

**Figure 2 nanomaterials-13-02948-f002:**
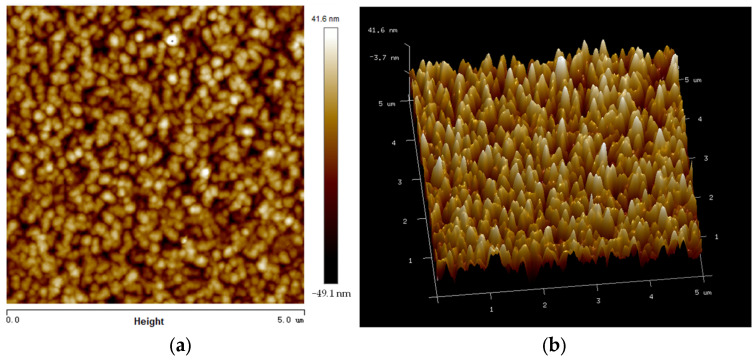
(**a**) Surface of the composite film by AFM; (**b**) 3D image of Fe_0.8_Ga_0.2_B film.

**Figure 3 nanomaterials-13-02948-f003:**
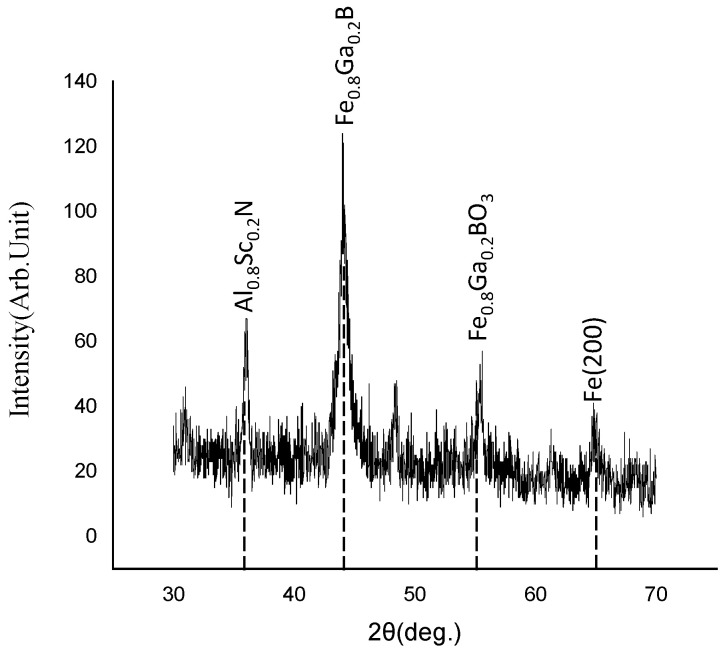
The XRD pattern of Fe_0.8_Ga_0.2_B film.

**Figure 4 nanomaterials-13-02948-f004:**
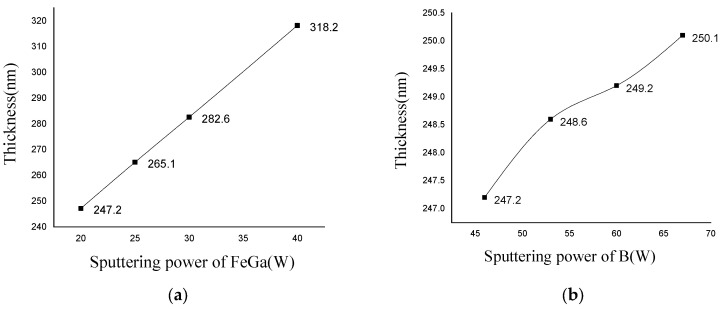
(**a**) Thickness variation for FeGa sputtering power; (**b**) thickness variation for B sputtering power.

**Figure 5 nanomaterials-13-02948-f005:**
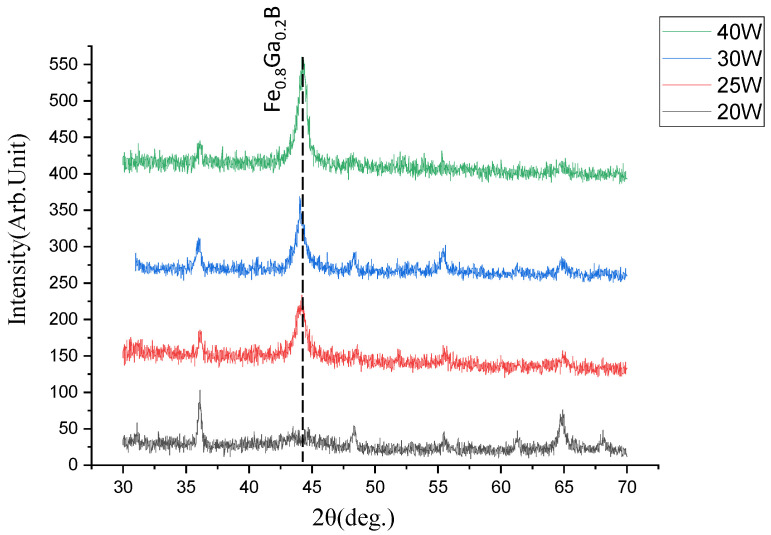
The XRD pattern of Fe_0.8_Ga_0.2_B film with different FeGa powers.

**Figure 6 nanomaterials-13-02948-f006:**
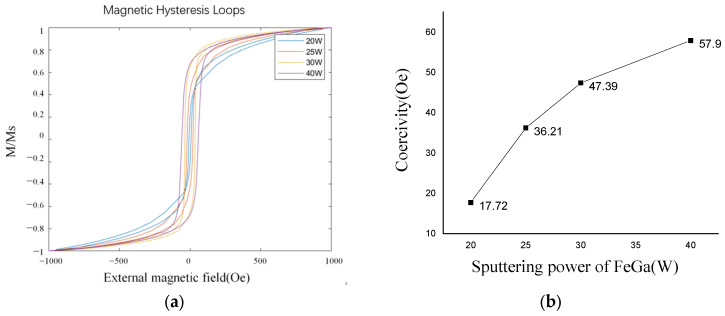
(**a**) Normalized (M/Ms) M-H loop of Fe_0.8_Ga_0.2_B films with different FeGa powers; (**b**) coercivity variation with FeGa sputtering power.

**Figure 7 nanomaterials-13-02948-f007:**
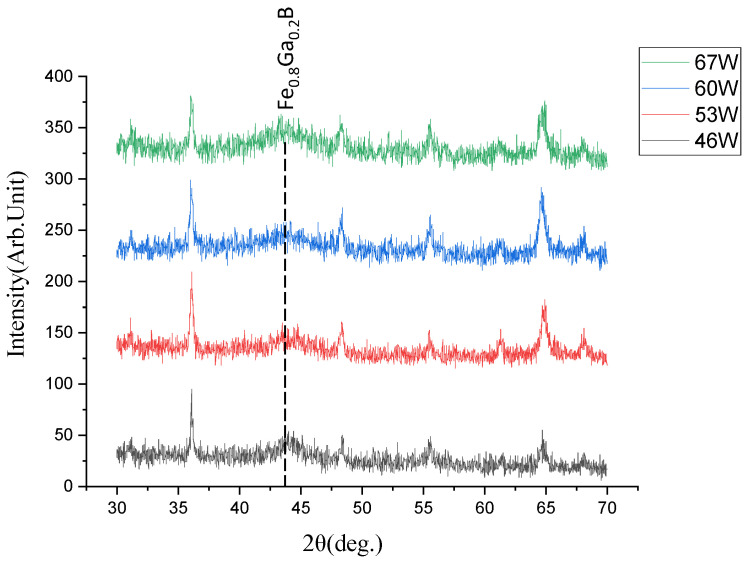
The XRD pattern of Fe_0.8_Ga_0.2_B film with different B powers.

**Figure 8 nanomaterials-13-02948-f008:**
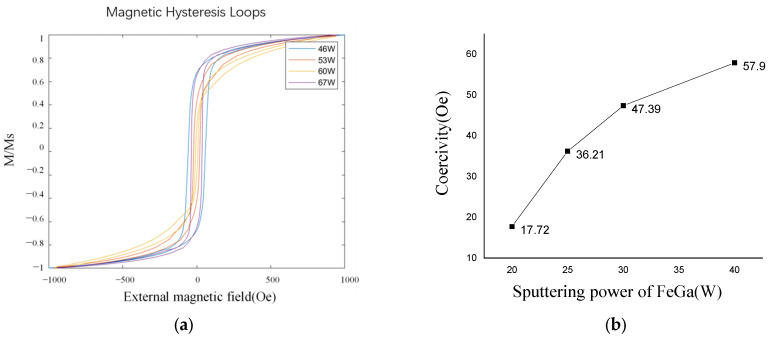
(**a**) Normalized (M/Ms) M-H loop of Fe_0.8_Ga_0.2_B films with different B powers; (**b**) coercivity variation with B sputtering power.

**Table 1 nanomaterials-13-02948-t001:** The grain size of the Fe_0.8_Ga_0.2_B thin film with different powers of FeGa target.

The Power of FeGa Target (W)	The Grain Size of the Fe_0.8_Ga_0.2_B Thin Film (nm)
40	11.8
30	8.2
25	7.0

## Data Availability

All data are contained within this article.

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
