# Peer review of "A Co-Sputtering Process Optimization for the Preparation of FeGaB Alloy Magnetostrictive Thin Films"

_nanomaterials, 2023, doi:10.3390/nano13222948_

Round 1
Reviewer 1 Report
Comments and Suggestions for Authors
The paper presents new results for FeGaB co-sputtering. The following points should be considered prior to publication.
Line 36: correct „graeter“
Line 41: what about CVD and electron beam evaporation?
Line 43: in my opinion, it should be “exhibit”
Line 50: In general, there exit many more magnetron sputtering approaches. I suggest to delete “generally” and to reformulate the sentence as: “There are two approaches employing magnetron sputtering using (i) a single cathode composed of FeGaB alloy or (ii) co-sputtering with two FeGa and B cathodes.”
Line 66: XRD, in which geometry, i.e., Bragg-Brentano?
Line 75: “A cross sectional diagram of the composite film is shown in Fig. 1.”
Line 76: delete “growth”
Lines 100 ff and Fig. 3: indicate peaks from different phases, i.e., which are the ones form AlScN and from FeGaB? What are the crystalline phases?
Lines 140 ff: a table of the obtained grain sizes should be added. Numbers have too many digits. For example: “11.76 nm” should be “11.8 nm” and “6.99 nm” should be “7.0” nm
Comments on the Quality of English LanguageMinor English revision recommended
Reviewer 2 Report
Comments and Suggestions for Authors
Introduction: I have no comments to add
Growth Process of FeGaB Magnetostrictive Films
Please decide FeGaB or Fe0.8Ga0.2B
Results and Discution
The authors discussed the research results in detail and conducted an in-depth discussion
The methodology of XRD, AFM and magnetic tests should be discussed in a separate chapter.
In the diffractograms of figures 3, 5, 7, it is necessary to specify which FeGeB planes the individual peaks correspond to.
